# Local Drug Delivery in Bladder Cancer: Advances of Nano/Micro/Macro-Scale Drug Delivery Systems

**DOI:** 10.3390/pharmaceutics15122724

**Published:** 2023-12-03

**Authors:** Irina V. Marchenko, Daria B. Trushina

**Affiliations:** 1Federal Scientific Research Center “Crystallography and Photonics”, Russian Academy of Sciences, 119333 Moscow, Russia; iramarchenko85@mail.ru; 2Institute of Molecular Theranostics, Sechenov First Moscow State Medical University, 119991 Moscow, Russia

**Keywords:** bladder cancer, non-muscle-invasive cancer, local intravesical drug delivery, colloidal drug delivery systems, indwelling devices, sustained delivery

## Abstract

Treatment of bladder cancer remains a critical unmet need and requires advanced approaches, particularly the development of local drug delivery systems. The physiology of the urinary bladder causes the main difficulties in the local treatment of bladder cancer: regular voiding prevents the maintenance of optimal concentration of the instilled drugs, while poor permeability of the urothelium limits the penetration of the drugs into the bladder wall. Therefore, great research efforts have been spent to overcome these hurdles, thereby improving the efficacy of available therapies. The explosive development of nanotechnology, polymer science, and related fields has contributed to the emergence of a number of nanostructured vehicles (nano- and micro-scale) applicable for intravesical drug delivery. Moreover, the engineering approach has facilitated the design of several macro-sized depot systems (centimeter scale) capable of remaining in the bladder for weeks and months. In this article, the main rationales and strategies for improved intravesical delivery are reviewed. Here, we focused on analysis of colloidal nano- and micro-sized drug carriers and indwelling macro-scale devices, which were evaluated for applicability in local therapy for bladder cancer in vivo.

## 1. Introduction

Bladder cancer is an uncontrolled growth of abnormal tissue that starts in the lining or connective tissue of the bladder (“non-muscle-invasive”) and can spread to the muscle wall (“muscle-invasive”). According to GLOBOCAN, it is the tenth most prevalent malignancy in the world and the thirteenth most lethal, claiming an estimated 212,536 lives in 2020 [1]. A comprehensive analysis of bladder cancer incidence and mortality worldwide, as well as a summary of demographic characteristics that relate to bladder cancer incidence in different countries, can be found in the latest review by L.M.C. van Hoogstraten et al., published in 2023 [2].

Schematic illustrations of different bladder wall components and cancer stages are presented in Figure 1. The unique anatomical properties of the bladder are governed by its multilayer structure. The wall of the bladder is composed of smooth muscle fibers oriented in multiple directions, known as the detrusor muscle. The inner aspect of a bladder wall is lined by the urothelium, with tight junctions making it difficult for conventional drug molecules to permeate through the bladder wall into tumor tissues. This stratified epithelium contains a layer of unique flattened “umbrella cells” whose surface area can increase when stretched. The impermeability of the urothelium is further enhanced by a very tight barrier of glycosaminoglycans, which form a mucin hydrophilic layer adherent to the luminal side. The glycosaminoglycan layer is very resistant to the passage of molecules and represents an important obstacle in the success of intravesical drug delivery.

The depth of invasion of cancer is a crucial factor in determining therapy and prognosis. When cancer spreads to the submucosa or lamina propria, it is considered a T1-stage cancer, defined as non-muscle-invasive cancer (NMIBC). NMIBC accounted for around 68–80% of all cases, with 42% in the Ta low-grade stage [1,3,4,5]. When cancer invades the detrusor muscle, it is a T2-stage cancer (muscle-invasive bladder cancer, MIBC). In this case, the gold standard of treatment is the radical cystectomy (removal of the urinary bladder). Finally, the T3 stage is when the tumor extends past the muscle into the perivesical fat, and the T4 stage occurs when the tumor spreads to nearby organs.

Approximately 25 to 32% of patients with bladder cancer present with MIBC (from which about 8% had T4), and despite a lower prevalence, the radical cystectomy raises a lot of concerns regarding the risk of cancer-related mortality and compromised quality of life [3,5]. Many NMIBC cases are suboptimally managed and are associated with a high recurrence (occurring in up to 80%) and progression rate. Moreover, 40–50% of cases progress to MIBC, which carries a 5-year survival rate of only 27–50% [5]. Effective treatment at the T1 stage will avoid removal of the bladder and improve the quality of life. This emphasizes the need for early diagnosis and appropriate primary treatment.

There are multiple therapies for T1 bladder cancer. Most of the work is devoted to the development of new drugs for bladder cancer which are administered in oral form, such as gefitinib [6], sunitinib [7], and nintedanib [8]. The main reasons for this bias are ease of administration and high patient compliance. The challenge consists in delivering an effective dose in the bladder together with minimization of the undesired biodistribution, which could increase side effects. Different strategies have been developed to improve drug efficacy and tolerability [5,9,10,11,12], and intravesical delivery can be considered as the preferred mode of drug delivery, although the pharmacodynamics of administered drugs is impaired by urinary production and excretion. Conventional treatment for NMIBC include (1) transurethral resection of bladder tumor (TURBT), in which malignant tissue is removed with an electrocautery device during cystoscopy; (2) intravesical immunotherapy with the Bacillus Calmette–Guérin (BCG) vaccine or chemotherapeutic drugs and surveillance. Bladder anatomy allows for relatively easy and direct delivery of drugs using a catheter or cystoscope through the urethra, allowing the drug to reach the targeted site with minimal side effects avoiding first-pass metabolism. Preliminary results of a Phase II study in 2023 showed that the use of neoadjuvant intravesical mitomycin C instillation immediately before operation appears feasible and demonstrates promising therapeutic efficacy in patients with NMIBC without significant adverse effects [13].

However, a number of obstacles must be overcome. First, drug dilution occurs as the urine is continuously produced and due to subsequent washout during urine voiding. To overcome this matter, different approaches including mucoadhesive drug delivery systems and pre-treatment with special agents can be implemented [9,14,15]. Potentially, drugs may be injected into the bladder wall by analogy with the intradetrusorial injection of a botulinum toxin, and this is generally accepted in the case of the lower urinary tract dysfunctions [16].

Second, the urothelium forms a low permeability barrier, allowing for poor penetration of drugs into the bladder wall [14]. It was shown that intravesical instillation results in 5–10% penetration of suramin into the bladder tissues, which is 2–3 times higher compared to mitomycin C, doxorubicin, and 5-fluorouridine [17]. Tissue concentrations declined log-linearly with respect to the depth of penetration, and a comparison of the concentration of mitomycin C in urine and immediately underneath the urothelium indicated a 10- to 30-fold decline in concentration across the urothelium [18,19]. Despite this, intravesical therapy of superficial bladder cancer with doxorubicin or mitomycin C instillations ensures that their concentrations in the urothelium and lamina propria tissues are at least 2000 times higher than with intravenous administration [20], which allows the significant dose reduction.

Third, repetitive/long-term catheterization may be associated with inflammation, bacterial infections, and discomfort, which reduce acceptance by patients. In this regard, the drug formulation for local delivery with the prolonged release, maintaining the required drug concentration, is in high demand.

The unique properties of the urinary bladder render it fertile ground for evaluating experimental approaches to local therapy. The extensive development of nanotechnology has led to the emergence of a wide variety of nanostructured systems, most of which have already been tested in vitro many times [21,22,23]. Due to the large gap between in vitro conditions and real body conditions, it is difficult to assess the potential of systems without experiments in vivo. Here, we focused only on those drug delivery systems and specific strategies for improving the efficacy of local delivery in bladder cancer that have already been evaluated in vivo and/or have undergone preclinical and clinical trials, advancing their transition from the lab bench into clinical practice. We give an overview of current and future prospects of colloidal nano- and micro-sized drug delivery systems, as well as indwelling three-dimensional depot systems for bladder cancer treatment. Novel drug delivery systems could increase the residence time of the drug in the bladder and have a potential to improve the efficacy of the therapy and therefore the survival rate.

## 2. Colloidal Nano- and Micro-Sized Delivery Systems

Substantial progress in the field of nanotechnology and the constant development and improvement of drug delivery systems have led to the emergence of many nanoplatforms for the diagnosis and treatment of bladder cancer. Micro-and nanocarriers for the delivery of chemotherapeutic agents (doxorubicin, cisplatin, pirarubicin, paclitaxel, and others), gene tools (small interfering ribonucleic acid, RNA), immunotherapy agents (BCG), and photodynamic drugs for bladder cancer are being developed [5,11,12,24,25].

To prolong the retention time and improve the permeability of chemotherapeutic drugs into the urothelium, mucoadhesive polymers have been widely developed as drug delivery systems. Among them, one of the most used is chitosan. Chitosan, a well-known mucoadhesive natural polymer, due to its amino groups, is able to adhere to negatively charged mucin, which is the main component of bladder mucosa. Chitosan nanoparticles help to overcome the bladder permeability barrier caused by a thick layer of mucus gel covering the bladder mucosa and lined with glycosaminoglycan chains dispersed in a viscous hydrogel. Chitosan nanoparticles are reported to be permeation enhancers through the mucus gel layer by accumulating in higher quantities at the glycosaminoglycan layer and forming a microconcentration gradient, resulting in increased diffusion of the drug into the bladder wall [26]. Concerning survival rates, instillation of BCG-loaded chitosan nanoparticles resulted in significantly longer survival than BCG commercial product (up to 86 days of survival with no systemic side effects). When compared to healthy bladder weight averages, all the groups (especially BCG commercial solution) showed higher bladder weights, confirming tumor formation. Histopathological findings confirmed antitumor activity in all treatment groups, and optimum findings were observed in groups treated with BCG-loaded chitosan nanoparticles. Liu et al. [27] developed paclitaxel-loaded chitosan nanosupension for sustained and prolonged delivery of paclitaxel. The formulation is formed as a result of the molecular self-assembly of paclitaxel and chitosan into nanofibers and the subsequent ultrasonic cutting of these nanofibers to obtain a nanosuspension. The paclitaxel/chitosan nanosupension has high paclitaxel loading efficiency, a positively charged surface, and good mucin adhesion, and it showed sustained release of paclitaxel. Histological analysis in mice after intravesical instillation showed that the tumor volume for the paclitaxel/chitosan blend was significantly larger than for the nanosupension. Also, the results of hematoxylin and eosin staining of tumors demonstrate that the necrosis areas in tumors treated with paclitaxel/chitosan nanosupension were larger, along with the disappearance of the nucleus. Successful tumor reduction was demonstrated using chitosan vehicles to deliver the reactive oxygen species-activated prodrug of gambogic acid in mice by Xu et al. [28]. In order to efficiently encapsulate hydrophobic payloads, part of the amino groups of low-molecular-weight chitosan was modified with hydrophobic moieties containing benzyl group. Benzyl alcohol was first reacted with 3,3′-dithiodipropionic acid to yield the glutathione-responsive intermediate. The cationic chitosan promoted the mucoadhesiveness (at least for 60 h after instillation) and penetrability (largest penetration depth was 200 μm) of a prodrug within the bladder wall. In vivo results also demonstrated no toxicity to normal urothelium [28]. In a later work [29], the authors successfully developed a co-delivery system consisting of a glutathione-responsive mucoadhesive nanocarrier based on a hydrophobically modified chitosan derivative for simultaneous delivery a highly potent NAD(P)H: quinone oxidoreductase 1 (NQO1) substrate KP372–1 and a reactive oxygen species (ROS)-activatable prodrug of epirubicin. After endocytosis, the promptly released KP372–1 could greatly increase the intracellular ROS level in a NQO1-dependent manner so that the epirubicin prodrug could be specifically activated inside cancer cells.

Moreover, a number of works demonstrate that coating with chitosan provide muchoadhesiveness of containers, e.g., based on poly-e-caprolactone [30] and nanodiamonds [31]. Martin et al. [32] modified the surface of poly (lactic-co-glycolic acid) (PLGA) nanoparticles with chitosan to increase transurothelial migration and tissue uptake. Relative to unmodified PLGA nanoparticles, the amount of fluorescence of a model dye was 9 and 14 times higher for nanoparticles modified with chitosan with molar weight 2.5 and 20 kDa, respectively, which demonstrates their higher uptake after instillation into the mouse bladder. A similar delivery system was developed in [33], where the authors showed that bladder retention was 6.5-fold higher for chitosan-coated PLGA nanoparticles than for uncoated ones.

However, chitosan has been reported to increase drug penetration into the pig urinary bladder wall only when high concentrations are used. In this case, deterioration and necrosis of the urothelium may arise and thus may limit the clinical application [34]. Incorporation of amine groups also provide mucoadhesiveness of the containers. Mugabe et al. [35] proposed a mucoadhesive formulation based on hyperbranched polyglycerols, which was hydrophobically derivatized with C8/C10 alkyl chains in the core and modified with methoxy-polyethylene glycol and amine groups in the shell for intravesical delivery of docetaxel. These carriers demonstrated more effective inhibition of tumor growth in an orthotopic model of bladder cancer compared to the commercial formulation of docetaxel. Another mucoadhesive docetaxel delivery system employed amine-functionalized polyacrylamide nanogels [36].

Incorporation of thiol groups also provide mucoadhesiveness to the PLGA containers [37]. Thiolated nanoparticles were able to form -S-S- bonds with cysteine-rich domains of the mucus glycoproteins, and such covalent bonds are stronger than the non-covalent interactions (e.g., van der Waals forces, hydrogen bonds, and ionic interactions with the anionic substructures of the mucus layer). Intravesical treatment with messenger RNA-loaded thiolated PLGA nanoparticles effectively inhibited the growth of orthotopic primary bladder tumors in mice and inhibited metastasis.

Recently, it has been discovered that fluorinated polymers could be utilized for the effective transmembrane transportation of biomacromolecules such as nucleic acids and proteins by utilizing the unique hydrophobic and lipophobic behaviors of fluorocarbon chains, with greatly enhanced efficiencies compared to their non-fluorinated counterparts. Fluorinated polyethylenimine (F-PEI) have been used as a carrier for the photodynamic agent chorin-E6 conjugated with catalase [38] and mitomycin C [39]. As shown, intravesical instillations of F-PEI-based nanoparticles greatly improved cross-membrane, transmucosal, and intratumoral drug delivery. Fluorinated chitosan was also shown to be an excellent transmucosal penetration enhancer [40,41,42,43]. In addition to catalase, nitazoxanide and hydrogenase modification of fluorinated chitosan nanoparticles was found to be beneficial in ameliorating hypoxia [41,42]. The role of catalase and nitazoxanide in fluorinated polyethyleneimine and fluorinated chitosan nanoparticles is to catalyze the in situ oxygen-generating reaction from endogenous tumor hydrogen peroxide, which dramatically boosted the potence of photodynamic or sonodynamic therapy [38,40,41].

Liposomes represent another type of nano-sized intravesical drug delivery system, alongside solid polymer nanoparticles [44,45,46]. Intravesical delivery of small activating RNA in liposomal formulation facilitates the expression of tumor suppressor in vivo, leading to regression and even disappearance of tumors in 40% of the treated mice [46]. Their surface is usually modified to impart targeting properties and enhance cell penetration. As shown, octaarginine surface modification of the BCG-loaded liposomes resulted in a decreased rate of tumor growth in rats [47]. Ph-responsive liposomes, upon exposure to acidic environment, underwent aggregation due to the loss of surface charge and associated with cells [45,48]. To impart mucoadhesiveness to conventional liposomes, they were decorated with polyethylenglycol (PEG) and maleimide-functionalized PEG [49]. Liposomes with maleimide groups exhibited superior ex vivo retention on the bladder tissue, which is related to their ability to form covalent bonds with thiols present in mucosal tissue. PEGylated liposomes were found to penetrate deeper into the mucosal tissue due to the stealth character of PEG that facilitates mucus-penetrating properties. The same research group demonstrated that maleimide-functionalized PLGA nanoparticles exhibited good mucoadhesive properties to the urinary bladder mucosa [50]. Regarding transfer to the clinic, Oefelein et al. [51] reported results of a Phase-I clinical trial of a third generation liposomal formulation of paclitaxel (TSD-001), specifically designed for NMIBC intravesical instillation (finished in 2020 [52]). TSD-001 delivers high urinary concentrations of paclitaxel with no measurable systemic exposure and is very well-tolerated in NMIBC patients.

One of the prospective carriers for intravesical therapy are nanotubes. Ex vivo mucoadhesion studies show that carbon nanotubes stick to the urothelium with a mean covering area of 5–10% [53]. In [54], the authors modified single-walled carbon nanotubes (SWNT) with phospholipid-branched polyethylene glycol and constructed a pirarubicin-loaded SWNT conjugate via a cleavable ester bond. Treatment with pirarubicin-SWNT demonstrated no side effects, unlike the pure pirabicine. Additionally, pirarubicin-SWNT exhibited higher tumor depression than pure drug (52.46 versus 96.85%). Yu et al. [55] demonstrated that synthetic chrysotile nanotubes loaded with circular RNA inhibited tumor growth and metastasis without obvious toxicity.

Microemulsions can be another promising option due to their high drug solubilizing capacity. Chen et al. [56] developed viscous microemulsions for co-delivery of cisplatin and gemcitabine as they produce a synergistic effect. The microemulsion form yielded an increase in the penetration depth in bladder tissue from 60 to 120 μm compared to the free drug solutions. Emulsion microgels stabilized with whey protein isolate showed good mucoadhesion to the bladder urothelium, and their retention in the bladder after intravesical instillation was observed for 24 h [57]. M.S. Saveleva et al. demonstrated that the local administration of microgels into the bladder significantly reduces their accumulation in other organs, avoiding side effects on healthy tissues, and provided better performance; the accumulation of the microgels in the bladder after intravesical instillation is almost 10 times higher compared to systemic intravenous administration [57].

Other types of carriers include protein nanoparticles. Gelatin nanoparticles loaded with paclitaxel provide pharmacologically active concentrations of the drug retained in tumors for at least 1 week [58]. Albumin-bound paclitaxel (Abraxane) exhibited minimal toxicity and systemic absorption in the first human intravesical Phase-I trial in 2011 [59]. In the Phase II trial in 2014, 36% of patients achieved a complete response after 6 weekly instillations of Abraxane [60]. The combination of albumin and paclitaxel molecules is beneficial for two reasons: it improves solubility of paclitaxel in aqueous environment, and additionally, it provides greater uptake of the drug in tumor cells owing to the albumin receptor-mediated transport. Albumin-bound rapamycin nanoparticles (ABI-009) are another potential targeted therapy for NMIBC [61]. In a Phase-1 trial (completed in 2016), intravesical ABI-009 exhibited minimal local toxicity and no systemic toxicity. Serum levels of rapamycin were achieved at all dose levels, indicating tissue penetration of the drug. Mullapudi et al. [40] developed a nanocarrier for gemtabicine delivery based on human serum albumin conjugated with a cancer-cell-targeting peptide. The carriers successfully reduced the tumor size (by 64–70%) compared to carriers without the peptide, whereas free gemtabicine was not effective in reducing the tumor burden.

Further promising carriers are polymer micelles. Jin et al. [62] used cationic 1,2-dioleoyl-3-trimethylammonium propane (DOTAP) to modify the methoxypoly (ethyleneglycol) (MPEG-PLA) nanoparticles. The prepared DOTAP-MPEG-PLA micelles showed 6-h retention in the bladder and enhanced drug permeability into the bladder compared to MPEG-PLA nanoparticles due to the positive charge that is capable of increasing the cellular uptake and tissue absorption of micelles. DOTAP-MPEG-PLA micelles loaded with doxorubicin showed significant inhibition of tumor size and repression of tumor weight in comparison with free doxorubicin. Hao et al. [63] used a tumor-targeting peptide to modify the containers, consisting of distearoyl hosphoethanolamine-poly (ethylene glycol) and platinum nanozyme loaded with the prodrug for the chemophotodynamic therapy of bladder cancer. The introduction of the platinum nanozyme was aimed to decompose H_2_O_2_ of tumor tissues into O_2_ to enhance the effect of photodynamic therapy. The authors demonstrated that this system enhances drug accumulation time and permeability to bladder cancer cells. Lin et al. [64] developed paclitaxel-loaded nano-scale-targeting micelles coated with a cancer-specific ligand named PLZ4, which are currently undergoing Phase-1 clinical trials [65].

High ability to penetrate the mucous membrane makes dendrimers another possible intravesical carrier. In [66] poly(amidoamine) (PAMAM), dendrimer has been conjugated with PEG to form the PEG–PAMAM complex for doxorubicin delivery. Instillation with doxorubicin-loaded PEG–PAMAM resulted in a higher penetration depth compared to free doxorubicin (860 μm versus 480 μm), as well as 10 times higher fluorescent intensity of doxorubicin in the bladder wall. Attenuation of tumor volume was observed in mice instilled with the doxorubicin-loaded PEG–PAMAM (15 mm^3^) compared to those treated with free doxorubicin (15 mm^3^ versus 75 mm^3^).

As was already mentioned, surface modification is a very popular strategy for improving tissue penetration. Surface modification of mesoporous silica nanoparticles with a second generation PAMAM dendrimers [67] or with thiols [68] ensured their enhanced mucoadhesiveness. Cell-penetrating polymer, e.g., poly(guanidinium oxanorbornene), improved the tissue penetration of modified PLGA nanoparticles by 10-fold in intravesically treated mouse bladder and ex vivo human ureter [69]. Poly(guanidinium oxanorbornene)-modified coumarine 6-loaded PLGA nanoparticles showed better urothelial and lamina propria penetration, which extended to the surface of the detrusor muscle, compared to unmodified nanoparticles or those modified with PEG, which did not penetrate into the lamina propria or the detrusor muscle.

As well as mucoadhesion, reduction responsiveness is another feature that helps the carrier to deliver a drug in bladder cancer cells. The intracellular glutathione level in tumor cells is as high as 2–20 mM, which is 100–1000 times and 7–10 times higher than that of extracellular matrix and normal tissues, respectively [70]. Guo at al. [71] developed a reduction-responsive cationic disulfide-crosslinked polypeptide nanogel of poly(l-lysine)–poly(l-phenylalanine-co-l-cystine) to deliver 10-hydroxycamptothecin (HCPT). This container possessed excellent stability under physiological conditions but rapid structural swelling and HCPT release in the reductive microenvironment. Poly(l-lysine) offered a positive charge to bond with the negatively charged bladder mucosa. Furthermore, the amphipathic nanogel with l-lysine residues allows HCPT to enter the cells in a similar way to amphipathic cell-penetrating peptides. Subsequently, the intracellular reductive conditions triggered the nanogel to deliver and release HCPT by cleavage of the disulfide bond. Treatment with the nanogel resulted in markedly higher HCPT fluorescence intensity in the bladder wall and induced a remarkable tumor growth inhibition compared to free HCPT treatment in both mouse [71] and rat [72] models. In [73], positively charged disulfide-crosslinked nanogel of oligoarginine-poly(ethylene glycol)–poly(L-phenylalanine-co-L-cystine) was found to prolong the retention period and enhance the penetration capability of the chemotherapeutic agent toward the bladder wall. PEG significantly improved the aqueous dispersibility of the HCPT-loaded nanogel and enhanced the mucoadhesive properties via the non-specific interaction between PEG chain and the bladder mucosa, accompanied with the electrostatic interaction between the cationic oligoarginine and the negatively charged mucosa. Moreover, oligoarginine, as a cell-penetrating peptide, efficiently penetrated through the cell membrane. The HCPT-loaded nanogel significantly enhanced tumor suppression in mouse and rat models [73].

Zhang et al. [74] used a very popular modern biological object—membrane nanovesicles obtained from bladder cancer cells. The authors shielded gemcitabine-loaded PLGA nanoparticles into nanovesicles, and the positively charged tumor-targeting hendeca-arginine peptide was used for surface functionalization to achieve targeting and mucus penetrating capacity. Compared to surface functionalization with free hendeca-arginine, surface functionalization with nanocomplexes greatly increased the amount of hendeca-arginine that remained on the cell membrane. These hendeca-arginine-modified carriers exhibited a higher antitumor effect compared to unmodified ones, which, in turn, had a higher effect than gemcitabine-loaded PLGA nanoparticles or pure gemcitabine.

One of the approaches is targeting fibronectin which is exposed on the surgical bed with residual tumor. Wang et al. [75] developed self-assembling peptide nanoparticles for doxorubicin delivery that target fibronectin. Upon binding, the nanoparticle simultaneously transforms into a fibrous coating which serves as a drug depot for the long-term release of doxorubicin. In vivo and ex vivo simulation of surgical injury on mouse urothelium showed that transformable nanoparticles provide an 8.5-fold higher doxorubicin concentration in urine after 3 days post-instillation and more intensive doxorubicin fluorescence in the targeted area compared to free doxorubicin.

An alternative approach which facilitates penetration of the drug into biological barriers is the fabrication of actively propelled containers. Choi et al. [76] developed a nanomotor which consisted of a polydopamine hollow nanoparticle with a urease-functionalized surface. This nanomotor was powered by urease to convert urea to carbon dioxide, and ammonia was assumed to be active in the urinary bladder, where urea is found in a significant amount. The authors demonstrated enhanced penetration to the bladder wall and prolonged retention in the bladder even after urination. Urease-modified polydopamine nanoparticles were observed in the bladder wall at a depth of 60 μm, which is deeper than for unmodified carriers.

### Additional Techniques to Improve Therapeutic Effect of Nano- and Micro Delivery Systems

Based on the above considerations, it becomes apparent that proper exposure of the bladder wall to the drugs is extremely challenging. The main problem in NMIBC treatment is the urothelium barrier, which has the highest transepithelial resistance among all epithelial membranes. In this context, several strategies can help to transfer drugs across the bladder, penetrating the barrier and improving delivery efficiency [14,77]. They can be categorized as chemical and physical affection. Using liposomes, cathionic polymers, fluorinated polymers, and organic solvents as chemical permeation enhancers and surfactants, discussed above, may be classified as chemical exposure. The use of dimethyl sulfoxide and protamine sulfate as the permeation enhancers are the most popular; acetone and ethanol are more rare [78,79]. These chemicals could irreversibly disrupt the barrier function of the urothelium and thus might lead to multiple side effects, such as bladder inflammation [80,81]. Moreover, previous studies have demonstrated that acetone has detrimental effects not only on urothelial cells but also on muscularis cells [82].

Among the physical approaches, the radiofrequency-induced thermochemotherapeutic effect, electromotive drug administration (iontophoresis/electrophoresis), and shock wave are noteworthy. Most often, these additional methods are used when the patient has high-risk NMIBC, unresponsive to BCG treatment. Radiofrequency-induced thermochemotherapeutic treatment involves the introduction of a mini-type antenna (915 MHz) in the catheter, which generates thermal energy to improve intravesical drug absorption [83,84,85]. The radiofrequency-induced thermochemotherapeutic mode of treatment may be an effective option for some patients who have experienced BCG failure and are not candidates for radical cystectomy, as well as an attractive alternative to BCG in the case of supply issues [84,86]. Van Valenberg et al. concluded that treatment with mitomycin C and simultaneous radiofrequency-induced hyperthermia of the bladder wall at 40.5–44 °C resulted in 10-fold higher drug concentration in the bladder tumor tissue versus cold mitomycin C instillation [87]. There is evidence that the frequency of recurrence-free survival in patients receiving intravesical chemotherapy combined with radiofrequency-induced hyperthermia was significantly higher than in patients who received BCG therapy [88,89]. Unfortunately, a Phase-III randomized control trial did not show a difference in bladder cancer outcomes between microwave-heated chemotherapy and standard treatment [90,91]. The up-to-date meta-analysis confirms this caveat that there is no statistical significance in preventing tumor progression [92].

Another physical approach employs an electrical current applied to the bladder wall, which enhances the transport of solubilized drugs across the tissue. For this, one electrode is inserted into the bladder via a spiral catheter, another placed on the skin of the lower abdomen, and an electrical current is passed between them. This electromotive technology improved the uptake of mitomycin C after instillation compared with passive transport [93,94], and a 5-year and 10-year cost-effectiveness study concluded that electromotive mitomycin C administration is a cost-effective treatment for patients with high-risk NMIBC [95]. Electromotive drug administration represents a viable option in patients with BCG-unresponsive NMIBC and reduces the risk of NMIBC recurrence [96,97]. A recent study comparing the efficacy of radiofrequency-induced mitomycin C therapy and electromotive mitomycin C administration showed no differences in the recurrence-free rate at 12 months, toxicity, and adverse effects [98]. Both methods may be considered as effective short-term alternatives to BCG in times of shortage. Even so, the systematic review based on the three trials with 672 participants stated that the possible impact of the use of radiofrequency-induced thermochemotherapeutic treatment or electromotive strategies on the development of side effects is still unclear and questionable and requires increased attention [99]. They cannot be recommended in the treatment of NMIBC due to insufficient clinical trial evidence. The pros and cons, including the estimated cost of the device-assisted NMIBC treatment and the results of clinical trials, have been very carefully analyzed by W.S. Tan and J.D. Kelly [84].

Recently, low-energy shock wave (which is the first-line treatment for renal calculi in most cases) has been reported to increase bladder barrier penetration through acoustic pulses [100] and thus to enhance intravesical epirubicin delivery into tumor tissues (by 5.7-fold) without subsequent toxicity [101]. Moreover, a preliminary pilot controlled clinical trial to establish the influence of a low-energy shock wave on NMIBC patients began in 2020 [102]. Bhandari et al. [103] proposed an ultrasound-guided drug delivery of cellulosic nanobubbles with encapsulated oxygen. The authors showed that oxygen nanobubbles can be propelled and precisely guided in vivo to the tumor by an ultrasound beam. Additionally, the ultrasound beam can also aid in oxygenation (this may be beneficial since the hypoxic cells are 2–3 times less responsive to therapy) and possible penetration of the bubbles into the tumor vasculature based on sonoporation. For intravesical therapy, nanobubbles were injected through a catheter into the bladder, and precise manipulation of the bubbles to hypoxic regions of the tumor was achieved by applying pulsed-wave Doppler ultrasound. Microscopy of cryosectioned tumor tissues showed that nanobubbles can be found at a depth of up to 500 μm inside the tumor. The therapeutic efficiency of simultaneously instilled mitomycin C was also increased based on a significant reduction in the rate of tumor progression using 50%-lower concentration of the chemotherapy drug. The drug delivery system with oxygen nanobubbles may be promising because of its multimodal (imaging and oxygen delivery) and multifunctional (targeting and hypoxia reduction) properties.

Control of delivery system localization by an external magnetic field may be an alternative to ultrasound-guided manipulation. The feasibility of using magnetic targeting was studied in vivo via instillation of magnetosensitive microparticles loaded with doxorubicin and application of a magnet placed on the skin surface at a predetermined site on the bladder [104]. The histopathological examination showed that the carriers were found within the bladder walls, predominantly at the targeted site where they were present at greater depths within the layers of the epithelium. Another study reported on epirubicin-loaded multiwall carbon nanotubes modified with magnetic iron oxide nanoparticles, which provided sustained release, prolonged retention behavior, and better antitumor activity of epirubicin compared to the free drug [105]. Immobilization of iron oxide magnetic nanoparticles on the surface of cells loaded with oncolytic adenovirus (capable of tumor-selective binding and killing) allowed for the anchoring of the “Trojan horse” closely to the targeted area, thus greatly enhancing their tissue penetration and anticancer efficacy in an orthotopic mouse bladder tumor model [106].

A schematic representation of the known techniques accompanying the intravesical administration of drugs in encapsulated form is summarized in Figure 2.

Summarizing the above, practically all the listed colloidal micro- and nanoformulations endow better penetration of the drug into the bladder tissue compared to drug instillation in the free form. The containers increase the retention time and penetration depth due to their mucoadhesive properties and functionalization with cell-penetrating molecules. For example, chitosan-based nanoparticles are detected in a bladder for 60 h after administration; gelatin nanoparticles and liposome in gel formulation provide retention of a drug in a bladder wall for 7 days. Chitosan-based nanoparticles penetrate the bladder wall to a depth of 1.5 mm.

In addition, nanocarriers can be modified with tumor-targeting molecules or provide the on-demand release of an encapsulated substance in response to the cancer cell environment. Among the nanoformulations, we believe that those that are currently undergoing clinical trials deserve special attention, namely, paclitaxel- and rapamycin-loaded albumin nanoparticles, liposomal formulation of paclitaxel, and PLZ4-coated paclitaxel-loaded nano-scale micelles. Strategies employing additional interventions to the bladder (including radiofrequency antenna of electrodes for electrophoresis) improve the effectiveness of instillation of colloid delivery systems mainly due to damage of the urothelium, which can have long-term side effects, and the feasibility of their use must be carefully weighed. The application of ultrasound and magnetically driven delivery systems may be promising, but they require prolonged exposure to fields and are still poorly studied. Despite all the efforts that are made to optimize nano- and microcontainers, their use is associated with a lot of limitations. The high recurrence rate of bladder cancer and the concomitant limitations of perfusion drugs require continuous improvement and optimization of the intravesical drug delivery systems. In particular, the development of long-term bladder retention systems is extremely relevant.

## 3. Reservoir-Type Intravesical Delivery Systems

Implantable delivery systems for intravesical therapy enable longer exposure of the urinary tract tissue to existing drugs, as compared to standard intravesical instillation and nano- and micro-sized delivery vehicles, as they remain attached to the bladder wall or floating in the bladder even after urine voiding. They can be categorized into degradable (e.g., gels) and non-degradable indwelling physical devices (e.g., pumps).

### 3.1. Biodegradable Systems

A novel approach is the use of hydrogels as a depot for the various formulations in the intravesical delivery [107]. Hydrogels are three-dimensional hydrophilic or amphiphilic polymer networks prepared via the formation of intermolecular bonds which can be chemical (covalently cross-linked networks) or physical (hydrophobic interactions, ionic or hydrogen bonds) in nature. Hydrogels can swell in water without disrupting their original structure and form an insoluble three-dimensional network with tunable degradability. Biodegradable polymer hydrogels provide a high concentration and a sustained release of the drugs at a tumor site, which eliminates the need for frequent drug administration [9,107,108,109,110,111]. Injectable physically crosslinked hydrogels have some advantages over chemically crosslinked formulations because they do not require photoirradiation, organic solvents, or crosslinking catalysts. In addition, physical crosslinking methods do not result in the production of heat during polymerization, which can affect the incorporated therapeutics, cells, and surrounding tissues. Physically cross-linked hydrogels are more readily eliminated after drug release and uptake into the urothelial tissues than hydrogels prepared using covalently bonded polymers. However, a balance is required between the prolonged duration of action and biodegradability so that covalently linked hydrogels do not cause any harm to the body. Among the wide variety of thermosensitive gels under investigation, only a few have been tested for intravesical delivery.

#### 3.1.1. Ion-Sensitive Formulations

The best-known examples of physically crosslinked hydrogels that can be gelled due to ionic interactions are polysaccharides, e.g., alginate, carrageenan, and chitosan. A significant advantage of polysaccharides lies in the possibility of their cross-linking at room temperature and physiological pH values. The mucoadhesive properties of polysaccharide carriers enable the desired attachment to the inner wall of the bladder and provide optimum release rate of the encapsulated agent. As regards in vivo-tested systems, calcium alginate rods containing mitomycin C (2 mm in diameter and 15 mm length in the dry form containing 0.2 mg of the drug) were implanted into the rabbit bladder [112]. Ultrasonographic observations showed that the alginate implant is retained at the injection site for 1 week, during which the drug is released fairly evenly, and 85% of the cargo drug was released at the end of the week. The authors observed calcification, congestion, and mixed-type inflammatory reaction, but those effects were minor and detected just around the implantation site of the alginate carriers. However, the essential fact is that alginate gels are destabilized during the extraction of calcium ions with chelating agents, which leads to their gradual dissolution.

Some classes of injectable hydrogel demonstrate a sol–gel phase transition upon injection in response to external stimuli such as temperature, pH, and light [108]. Among the different types that have been developed, thermosensitive hydrogels have gained increasing attention [108,109,113].

#### 3.1.2. Thermosensitive Formulations

Thermosensitive hydrogels are formed by chemical cross-linking using a covalent bond between polymer chains or hydrophobic reactions. Chitosan is a very attractive cationic polymer because of its biocompatibility with negligible immunostimulatory activities. Moreover, positively charged polymers can alter the penetration of drugs into the bladder wall because of the electrostatic interaction with the negatively charged glycosaminoglycan layer of the bladder. To control the chitosan solubility through hydrophobic interactions and hydrogen bonds, β-glycerophosphate (approved by the FDA for intravenous administration) was proposed. Chitosan/β-glycerophosphate mixtures are transparent below physiological temperature due to electrostatic attraction between phosphate groups of β-glycerophosphate and ammonium groups of chitosan; moreover, the additional hydration due to the hydrogen bonding of glycerophosphate with water molecules prevent gel formation. The sol–gel transformation occurs upon heating to 37 °C, and the gelation can be attributed to a thermally induced shift in interchain forces of attraction compared to repulsion, which may arise through several mechanisms [114]. Generally, glycerol provides a protective and hydroresistant layer around the chitosan chains, and increasing the temperature eliminates this polyol layer and allows the chitosan to be in equilibrium through stronger hydrophobic bonds, thereby generating gels. The rheological data, mucoadhesion and retention on the porcine bladder mucosa, syringeability through the urethral catheter, and the mitomycin C release profiles (up to 39% of the mitomycin C were released in 6 h from a gel) reveal that chitosan with the molecular weight of 370 kDa, combined with β-glycerophosphate, exhibited excellent resistance to urine wash-out [115].

The main drawback of the chitosan/β-glycerophosphate gels is that the β-glycerophosphate impairs the mucoadhesive properties of chitosan (which is related to the reduction in positive values of zeta potential for these formulations compared to chitosan alone). So, in terms of the bladder retention of the formulation, additional improvements may include the introduction of magnetic nanoparticles into the chitosan/β-glycerophosphate gel. It was shown that chitosan/glycerophosphate/Fe_3_O_4_ with encapsulated Bacillus Calmettee–Guérin (BCG) or mitomycin C adhered to the bladder walls and resisted being washed away during urine voiding, and the gel system can withstand the hostile environment of rat urinary bladder for a limited period of 48 [116] or 72 h [117], correspondingly. The antitumor effect and increased survival rates after intravesical mitomycin C delivery by chitosan/glycerophosphate/Fe_3_O_4_ hydrogel were registered in [117]. The uptake of mitomycin C through the bladder mucosa increases from 7.80 ± 0.46 to 21.25 ± 1.55 μg in the case of delivery using chitosan/glycerophosphate/Fe_3_O_4_ gel compared with the free substance [117].

Alongside magnetically driven localization, an additional option for preventing urinary tract obstruction is related to the floating hydrogel [109]. It was shown that an in situ gelling system based on thermosensitive Poloxamer 407 and NaHCO_3_, loaded with adriamycin or gemcitabine, can float owing to the decomposition of NaHCO_3_ in the presence of H^+^ [118,119]. The production of carbon dioxide that is attached to the surface of hydrogel allows the material to float in a model environment (Figure 3A). Figure 3A comprises a sequence of photographs demonstrating the introduction of a liquid substance into a model medium, simulating the introduction of a gel into the bladder in vivo. Photos 1–3 demonstrate the immediate formation of a hydrogel from the liquid mixture containing 35% Poloxamer 407 and 8% NaHCO_3_. In the third photograph, the catheter responsible for introducing the mixture was removed, and its supply was terminated. Meanwhile, NaHCO_3_ undergoes decomposition in acidic environments, such as citric acid buffer and acidified urine, leading to the production of numerous CO_2_ bubbles (Photo 4). The adriamycin-loaded hydrogel is further supported by microbubbles which are generated on its surface and within it. This contributes to its fast ascent (Photos 5–6) and acts as evidence that it would not cause bladder obstruction.

It has been mentioned that to achieve the excellent floating and bladder retention effect, the urine needed to be acidified, which makes the approach less acceptable due to the potential irritation of the bladder caused by low pH. To address this issue, an updated version of the intravesical floating gel contained only pure P407 (without NaHCO_3_) is administered to the animals in an oxygenated state [120]. Oxygenated-by-shaking concentrated poloxamer solution is saturated with self-generating microbubbles, which can be viewed as gas-filled micelles formed by surfactant molecules, whose hydrophobic tail groups face the hydrophobic gas, and whose hydrophilic head groups face the aqueous phase. Due to the viscosity of P407 solution, microbubbles were suspended for a certain amount of time and did not escape easily. In vivo release experiments showed that the drug was released continually from hydrogel for 10 h during the lengthened dissolution process comparatively to P407/NaHCO_3_ composition [120]. In a bladder simulation model that involves emptying the bladder every 2 h, the remaining fraction of the P407 gel decreased by approximately 15% at every pouring point, with a reduction of approximately 60% at 8 h, and erosion followed zero order kinetics [121].

Another approach to making the Poloxamer 407-based floating gel is to add perfluoropentane to its composition [122]. The ultrasonically emulsified mixture of Poloxamer 407-perfluoropentane was administered intravesically, and since the boiling point of perfluoropentane is 29 °C, it evaporated in the rabbit bladder to form microbubbles in the hydrogel. Furthermore, perfluoropentane is biologically inert and clinically approved as an ultrasound contrast agent, which allows the gel to be monitored in vivo (Figure 3B).

**Figure 3 pharmaceutics-15-02724-f003:**
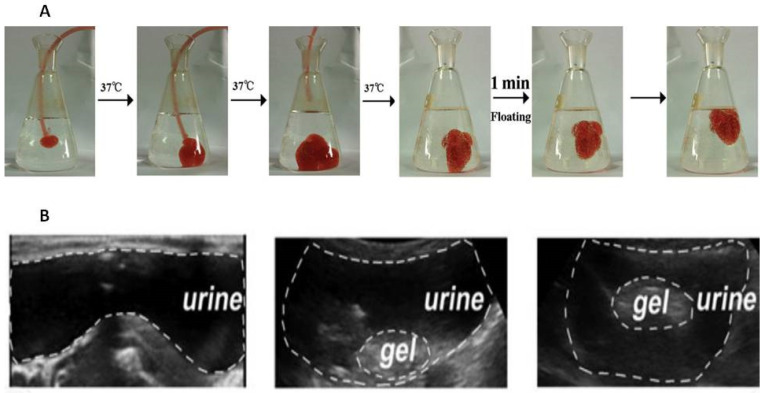
Adriamycin-loaded Poloxamer 407-NaHCO_3_ floating hydrogel with the peripheral microbubbles (**A**) [118]. Poloxamer 407-perfluoropentane hydrogel floating in the rabbit bladder after 2 min after injection (**B**) [122].

It has been established that the gel formulation containing F127 and deguelin-loaded nanoparticles was syringeable at room temperature, and the residence time of the drug in the bladder was maintained for 6 h [123]. When comparing gels based on chitosan and Poloxamer 188/407, samples based on chitosan showed the best performance in terms of their stability in the Tyrode solution, which mimics the conditions in the bladder (Poloxamer gels lost their in situ gelling properties at body temperature), bioadhesion to the bladder mucosa, and the percentage of the cargo drug gemcitabine permeating the bladder mucosa (the amount of gemcitabine was almost twice as high in case of chitosan hydrogel) [124].

Biodegradable poly(N-isopropylacrylamide) (PNIPAM) is another well-known thermosensitive polymer with a gelation temperature of 32 °C, which serves as a base for intravesical degradable formulations [110]. A comparison of the amount of cisplatin accumulated in bladder tissues after administration in the form of its solution and in the composition of PNIPAM or PNIPAM grafted sequentially with hyaluronic acid and gelatin (PNIPAM-HA-G) gels was made [125]. In vivo results showed a seven-fold and two-fold benefit after 6 h exposure when using the PNIPAM or PNIPAM-HA-G depot systems correspondingly, presumably because of the mucoadhesiveness of the hydrogels (owing to cationic nature of PNIPAM). Although histological examination showed no adverse change in the urothelium, PNIPAM caused partial desquamation of umbrella cells [125]. Unfortunately, there is information that monomeric acrylamide, which is a main metabolite of PNIPAM in the body, exhibits carcinogenic or teratogenic toxicity [126], which sufficiently limits the clinical application of PNIPAM.

Modified thermosensitive triblock co-polymer polyethylene glycol-poly[lactic acid-co-glycolic acid]-polyethylene glycol (PEG-PLGA-PEG) is the another type of in situ-formed gel [113,127]. Gelation at body temperature extended the residence time of the cargo molecules in the bladder from 8 to 24 h, while additionally, the emollient properties of the gel were helpful [113]. Moreover, thiol-bearing 2-(acetylthio)ethylacrylate (ATEA)-based gels showed good intravesical retention and helped to resist the washout of encapsulated doxorubicin molecules due to covalent disulphide bridges with the cysteine-rich regions of urothelial mucins [128]. In general, mucoadhesiveness and modification with cationic groups (-amine, -thiol, etc.) are often used to increase the bioavailability and duration of action of intravesical drugs [109].

Mini-tablets, extrudates, and mini-molds with a lipid matrix can be profitable dosage forms for long-term intravesical treatment. As the hydrolysis of lipids would not occur in the urinary bladder, the lipid-based formulations would maintain the integrity for a long time and provide the long-term control-released properties in the bladder. In addition, the density of the lipids was lower than urine. For this reason, different small-sized glyceryl tristearate-based dosage forms were prepared to permit application through the urethra [129,130]. The release kinetics depend on the shape of the formulation: spherical tablets with a diameter of 2 mm provide a five-day release; while 4 mm-diameter tablets allow for almost five-times-slower release [129]. In vivo evaluation showed that the prepared long-term floating preparations could maintain an effective 5-fluorouracil concentration in the bladder for about one month; furthermore, in this period, the 5-fluorouracil concentration in blood was always far less than that in urine [130].

#### 3.1.3. Combined Particle–Hydrogel Systems for Intravesical Delivery

To enhance stability of nanoparticles and to minimize their susceptibility to washout, different combined systems where nanoparticles are incorporated in a hydrogel have been developed. Men et al. [123] designed a delivery system for deguelin, which is poorly water soluble. For this, deguelin was encapsulated in cationic DOTAP and monomethoxy poly(ethylene glycol)–poly(3-caprolactone) hybrid nanoparticles, with subsequent dispersion into a thermosensitive Pluronic F127 hydrogel. The hydrogel enhances the tissue absorption and cellular uptake of the nanoparticles and prevents their elimination during urination. GuhaSahar et al. [131] developed in situ a gelling liposome-in-gel system composed of fluidizing liposomes incorporated into a urine-triggered hydrogel. The liposomes enhance cellular penetration through the urothelial barrier, while the hydrogel co-delivers the suspended liposomes and enhances adhesion on the mucin layer of the urothelium. The paclitaxel-loaded liposome-in-gel system showed drug retention for at least 7 days, which is higher than the free drug approach (a few hours). Another liposome-in-gel system was proposed in [132], wherein the rapamycin-loaded folate-modified liposomes were dispersed in the same Pluronic F127 hydrogel. The gel with folate-modified liposomes loaded with rapamycin showed higher inhibition of tumor growth compared to unmodified liposomes. Karavana et al. [133] prepared bioadhesive gemtabicine-loaded microspheres from Carbopol 2020 NF and Eudragit E100 and dispersed them in two gel formulations: mucoadhesive chitosan gel and in situ Poloxamer gel. Intravesical treatment with a once-weekly Poloxamer-based delivery system was found to be more effective than the chitosan-based system and the delivery of gemtabicine-loaded microspheres without a gel. Unfortunately, this result is blurred by the fact that upon the dilution with an artificial urine, the Poloxamer-based gel lost its in situ gelling properties at body temperature.

As for the gels that have reached clinical trials, several potential thermosensitive gel systems should be noted. TCGel^®^ is a hydrogel with reverse thermal gelation properties, produced by TheraCoat Ltd. (Raanana, Israel), containing Pluronic F-127, PEG-400, and a small amount of hydroxypropyl methylcellulose. Following instillation, the gel solidifies and acts as a sustained drug release depot in situ. It exhibited improved safety and residence within the bladder cavity between 6 and 8 h with the release of mitomycin C during this time [9,134]. TCGel^®^ is slowly excreted from the bladder during urination. It is 100% biocompatible and harmless to the body. TheraCoat Ltd. has commenced several efficacy clinical trials [135,136].

UGN-102 is another investigational agent designed for primary non-surgical treatment of NMIBC and to potentially obviate the need for repetitive TURBT. UGN-102 consists of mitomycin and a proprietary reverse thermal hydrogel (UroGen Pharma, Raanana, Israel). The ablative effect of UGN-102 was evaluated after six intravesical once-weekly instillations of UGN-102 [134]. The gel slowly disintegrates over a 6 h period and is eliminated through normal urine flow, allowing for the sustained release of mitomycin over a period of 4 to 6 h, enabling the ten-fold-higher drug content in the bladder after 2 and 6 h after instillation as compared to the free drug [137]. At the same time, levels of mitomycin C in the plasma were low, confirming the safety of the treatment. The achieved disease-free level was 95, 73, and 25% after 6, 9, and 12 months after treatment initiation, respectively [134]. Assuming positive findings from the ENVISION [138] and ATLAS [139] Phase-III studies in the summer of 2023, UroGen anticipates submitting a new drug application for UGN-102 in 2024. If approved, UGN-102 would be the first non-surgical primary therapeutic to treat a subset of bladder cancer characterized by high recurrence rates and multiple surgeries.

In addition, there is an intermediate product worth mentioning in this section—an intravesical delivery system composed of drug-encapsulating biodegradable polycaprolactone (PCL) microcapsules and connected by a bioabsorbable Polydioxanone (PDS) suture with NdFeB magnets in the end (Figure 4A,B). The implant can be easily inserted into the bladder and forms a “ring” shape under the magnet exposure (Figure 4C,D). In this study, inserted devices were retained in a swine model for 4 weeks. The partially resorbable implant enables the controlled release of therapeutic agents without burst release, which is ensured via long-term retention in the bladder [140]. After a month, single polycaprolactone microcapsules were found in the bladder (Figure 4E), which were quickly excreted from the body within 48 h.

### 3.2. Non-Biodegradable Indwelling Devices

#### 3.2.1. Elastomer-Based Devices

The mechanical properties of elastomers can be advantageously used for the insertion and retention of products based on them in hollow organs. But in contrast to biodegradable products, their significant drawback is the need for a transurethral removal procedure, which inevitably reduces the patient’s compliance. More recently, over the past few years, bladder devices based on elastic polymers (elastic resin) have been fabricated via stereolithographical technology to achieve the desired complex geometrical structures, shape, and mechanical properties of the implant to enable its insertion via catheter [141]. A modification of the 3D printing process allows for the formation of two variants: hollow and solid implants with a diameter of 3 mm and a length of about 130 mm in an extended shape (Figure 4F–K). This option allows for the adjustment of the release duration of the cargo molecules from the bladder-retentive devices in the range of 4–14 days [141]. Additionally, this technique can be used to print drug-containing devices regardless of their solubility in water as they can either be dissolved or dispersed in the liquid resin.

A resorbable elastomer with poly(glycerol-co-sebacic acid) composition [142] may be of interest for intravesical delivery owing to better biocompatibility than poly(DL-lactide-co-glycolide) when tested in vivo [143]. Unlike poly(DL-lactide-co-glycolide), poly(glycerol-co-sebacic acid) primarily degrades via surface erosion, which gives a linear degradation profile of mass, preservation of geometry and intact surface, and retention of mechanical strength. A poly(glycerol-co-sebacic acid) tube with a laser-drilled orifice (Figure 4L) allowed for the drug payload release via osmotically-driven water permeation over a time period of a few weeks [142], and modulation of the release rate can be achieved by varying the orifice size. In vitro experiments have shown that the elastomer is susceptible to hydrolytic degradation, indicating the possibility of creating a completely resorbable device.

A number of semi-solid printing implants were fabricated via pressure-assisted micro syringe printing based on polycaprolactone and ethylene vinyl acetate copolymer [144]. They were flexible enough to be inserted using a common urinary catheter (Figure 4M) and remain inside the urinary bladder for up to several weeks. 

**Figure 4 pharmaceutics-15-02724-f004:**
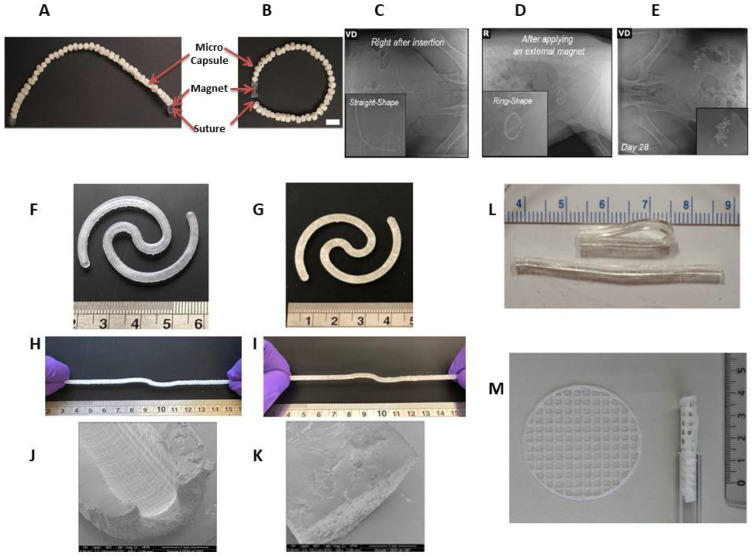
Optical image of a partially biodegradable ring-shaped implantable device based on polycaprolactone, polydioxanone, and NdFeB magnets before (**A**) and after (**B**) exposure to a magnet. Scale bar = 1 cm. Radiograph images of the implant immediately after insertion (**C**), after applying an external magnet (**D**), and 28 days after implantation in a swine model (**E**) [140]. Photographs of hollow (**F**) and solid (**G**) bladder devices (elastic resin) in their intact form and under stretching (**H**,**I**). Scale is in cm. SEM images of sections of the empty hollow (**J**) and empty solid (**K**) devices [141]. Poly(glycerol-co-sebacic acid) tubes demonstrating flexibility of material (**L**) [142]. Printed net-shaped polycaprolactone/ethylene vinyl acetate copolymer implants unfolded (left) and coiled up (right) for insertion via catheter [144] (**M**).

#### 3.2.2. Osmotic Pumps

Among non-degradable implantable systems that provide long-term drug delivery with a minimal risk of infection and minimal risk to patient’s life, a number of state-of-the-art osmotic devices should be noted, e.g., LiRIS, UROS, GemRIS, and Duros. 

A continuous lidocaine-releasing intravesical system LiRIS (TARIS Biomedical, Lexington, USA) is the only indwelling bladder device that has advanced to clinical trials [145,146]. Prior to LiRIS, an intravesical horseshoe-shaped device, UROS (Situs Corporation, Houston, TX, USA), was developed (Figure 5A,B) [147,148]. This indwelling pump was 10–15 cm in the largest diameter and released cargo for up to 28 days [149]. Phase I/II-trials were relatively unsuccessful due to the poor tolerability associated with the large size of the device; thus, UROS has not entered clinical practice [149]. Thereafter, LiRIS was developed in order to minimalize the size to avoid discomfort [145]. LiRIS is a dual-lumen silicone tube that contains the drug in the form of mini-tablets in one lumen (enabling a higher dose of 2 mg) and a super elastic shape memory nitinol wire in the other (Figure 5C). To enable intravesical administration, the wire is mechanically forced into an elongated shape. The device can be inserted into the bladder with a foley catheter and adopts a “pretzel” conformation once inside the bladder, preventing it from being accidentally voided. A silicone container absorbs urine to dissolve the lidocaine contents, whereas the created osmotic pressure forces the solution out of the container through a small orifice in a controlled release over 14 days [145,150].

Promising results were achieved when the device was tried in a rabbit model (Figure 5D); lidocaine was detected in the bladder tissue during the 3-day period, while a single instillation yielded immeasurable amounts within 24 h [150]. The small proof of concept study in women with ulcerative interstitial cystitis and Hunner′s lesions demonstrated a favorable safety profile as well as long-lasting improvements in lesions, pain, and voiding frequency after 2 weeks of LiRIS therapy [151]. Expanded Phase-II/III trials conducted in 2021 to evaluate the efficacy of a LiRIS device in female patients did not demonstrate a therapeutic effect of LiRIS compared to placebo [146].

The upgraded version of the silicone “pretzel” pump with a shape memory wire is adapted to gemcitabine bladder delivery (GemRIS or TAR-200, TARIS Biomedical) [152]. The device consists of a 5 cm semipermeable silicone tube that slowly releases dissolving gemcitabine tablets (Figure 5E,F), and as a result, 60–70% of the cargo drug is delivered over 2 weeks, compared to the 2 h conventional dwell time for intravesical drugs [152]. GemRIS proved its safety and tolerability during the 7-day indwelling time and demonstrated encouraging preliminary efficacy [153]. It showed very promising results during Phase-I trials completed in 2019–2020 for single-agent delivery in patients with cisplatin-ineligible muscle-invasive bladder cancer [154,155,156]. Along with safety and tolerability, the aim of a Phase-I study was to unleash the potential of TAR-200 for intravesical drug delivery in muscle-invasive bladder cancer in the neoadjuvant setting in combination with nivolumab [157]. A currently ongoing Phase-III trial of combination treatment with intravesical TAR-200 and systemic cetrelimab, with the last update in August 2023, aims to evaluate the efficacy in participants with muscle-invasive urothelial carcinoma of the bladder [158]. In a recruiting Phase-III study started at the beginning of 2023, metronomic dosing of intravesical gemcitabine, delivered via TAR-200 (alone or in combination with cetrelimab), will be evaluated and compared against intravesical Bacillus Calmette–Guérin in participants with high-risk NMIBC [159]. A few years earlier, the FDA granted fast track designation to TARIS Biomedical for GemRIS (TAR-200) for the treatment of patients with muscle-invasive bladder cancer [160].

A completely different type of implantable device consists of a cylindrical titanium alloy drug reservoir (4 mm in diameter by 45 mm in length and holds approximately 150 µL of formulation), capped at one end by a polyurethane rate-controlling semipermeable membrane (Figure 5F). The osmotic engine consists mainly of NaCI adjacent to the membrane, and a sliding elastomeric piston isolates the engine from the drug formulation. At the far end of the cylinder is the diffusion moderator that contains the orifice through which the drug is released. The DUROS (ALZA Corporation, Vacaville, USA) osmotic pump has been commercialized for the subcutaneous implantation and palliative treatment of prostate cancer (Viadur system) [161]. This implant provides zero-order release kinetics [161,162] and can be adapted to biomolecules delivery that requires long-term controlled administration, including those that have a narrow therapeutic window or a short half-life. DUROS delivery technology is capable of delivering a wide range of therapeutic peptides and proteins (1.2 kDa–25 kDa) for durations ranging from 3 to 12 months [161,163] and may be applicable to the chronic delivery. Several facts, namely, that the stability of the peptide molecules was maintained for 3 years at 37 °C [163], its proven safety for more than a year after implantation in dogs [164], and the favorable feedback from patients in clinical studies of a 1-year DUROS device [165], gives us reason to consider the implant applicable for bladder cancer treatment.

**Figure 5 pharmaceutics-15-02724-f005:**
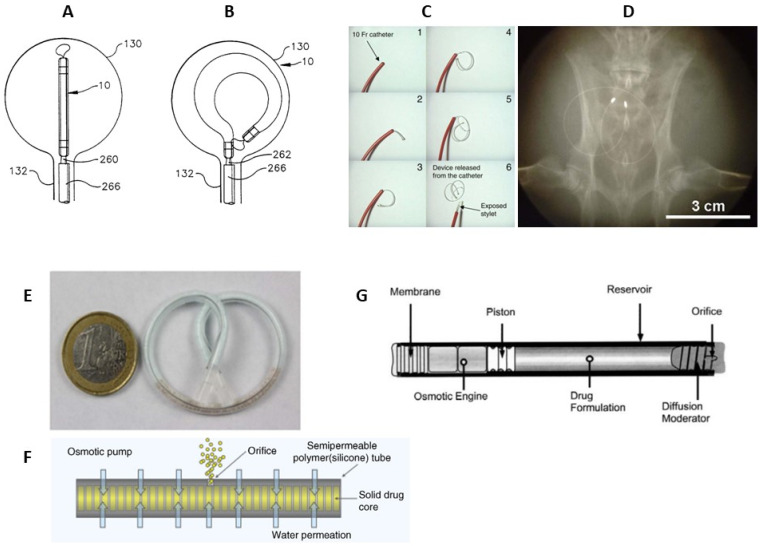
Scheme of the UROS device during insertion into the bladder through a catheter (**A**) and after taking on a horseshoe shape (**B**) [147]. The representation of deployment of the LiRIS by the catheter–stylet system (**C**). X-ray image of an implanted device within the bladder of a rabbit in a ventral–dorsal view (**D**) [150]. GemRIS (TAR-200) device after self-coiling into a pretzel shape (**E**) and its scheme (**F**) [152]. Sketch of DUROS implant (**G**) [161].

## 4. Discussion

In recent years, numerous studies have been carried out to overcome the shortcomings of the traditional intravesical instillation of therapeutic solutions. There are two main challenges associated with drug delivery to bladder cancer that are being addressed: poor drug bioavailability due to the barrier properties of the urothelium; and a gradual decrease in drug concentrations due to periodic urination. Various nano- and micro drug delivery systems, such as liposomes, micelles, dendrimers, nanogels, and polymer nanoparticles, have been examined to increase the bioavailability of the encapsulated drug, to ensure patient adherence to the treatment, and to reduce drug-related toxicity.

Concerning nano- and micro-sized colloidal systems, almost all of them are designed to possess mucoadhesive properties, which prolong the residence time in the urinary bladder from several hours to days compared to instillations of the drug solution in free form. In this context, chitosan and gelatin nanoparticles, as well as combined liposome-in–gel formulations, deserve attention. Co-administration of permeation enhancers, or supplementing intravesical chemotherapy with physical interventions (such as radiofrequency-induced hyperthermia, electrophoresis, and shock wave), increases the depth of drug penetration into the bladder tissue, but this is mainly due to destruction of the urothelial barrier, which can have long-term side effects. At the same time, the use of ultrasonic influence and a magnetic field to manipulate the localization of colloidal systems can be very useful; through external influence, with minimal side effects, the required amount of medicine is maintained in the targeted area. The main issue that remains unresolved is the fairly rapid excretion of colloidal drug delivery systems from the body which occurs within a few days (with a maximum of a week), which necessitates undesired repeated catheterization.

Vesical retention of the macro-scale indwelling reservoirs may be limited by the rate of bioresorption in the case of biodegradable systems such as thermo- and ion-responsive gels (within a maximum of several weeks). In the case of non-degradable intravesical devices, the duration of their meaningful stay in the bladder is determined by the amount of drug they carry. Such devices maintain the concentration of the drug for a much longer period (for months), but after their use, they may need to be removed from the body. Many non-degradable macro-scale devices are undergoing clinical trials, and among them, GemRIS implant shows great promise. With the appropriate choice of therapeutic molecules, “pretzel” pumps may have potential in bladder cancer treatment. We are driven by the following considerations: (1) the reservoir size could be balanced so that it is not too small to be voided out yet not too large to cause bladder irritation or obstruction; (2) the reservoir could be loaded with a sufficient amount of cargo drug and should be able to implement constant release of a precise drug quantity into the bladder for at least 1 month.

## 5. Conclusions

The development of bladder drug delivery systems for cancer treatment is a very dynamic field. Ten years ago, only single formulations reached in vivo testing; now, there are more than a dozen of them. Promising solutions exist among all types of intravesical delivery systems. Among those that have successfully passed clinical trials and could potentially enter clinical practice, reverse thermal hydrogel UGN-102 (UroGen Pharma, Raanana, Israel) and “pretzel” pump GemRIS or TAR-200 (TARIS Biomedical, Lexington, KY, USA) are worth noting.

In our opinion, the path to creating an optimal intravesical formulation is to couple the significant benefit of in situ gelation within the bladder following instillation (when the hydrogel solidifies and acts as a sustained drug release depot in situ) with the ability to adjust the degradation time of the hydrogels (to prolong their residence in the bladder). To summarize, the potential optimal formulation will combine (1) the need for a single dose with ease of administration (as a solution through a catheter); (2) significant content of the drug component, which will be gradually released during hydrogel degradation; (3) long-term retention in the bladder without affecting the outflow of urine; (4) gradual degradation with complete self-elimination from the body. Successful sustained intravesical drug delivery can eliminate the need for multiple catheterizations, leading to improved patient compliance, so it is important for any form of therapy.

## Figures and Tables

**Figure 1 pharmaceutics-15-02724-f001:**
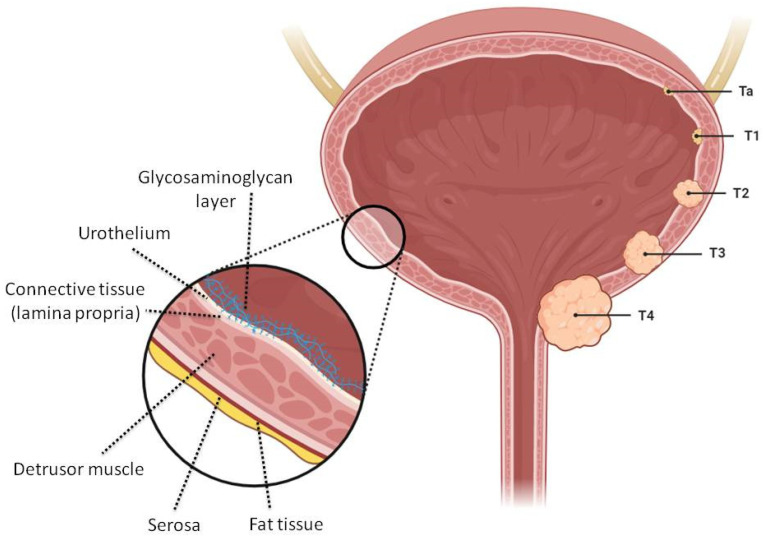
Anatomical illustration of different bladder cancer stages and a zoomed-in bladder wall with its contents. The types and stages of bladder cancer: Ta and T1 types are confined to the mucosa. The second stage (T2) invades the muscle layers either superficially or deeply. After penetration into the muscle and reaching the fatty tissue, the T3 stage is reached, where T4 is characterized by the invasion of surrounding glands such as the prostate, uterus, bowel. Created with BioRender.com (10 October 2023).

**Figure 2 pharmaceutics-15-02724-f002:**
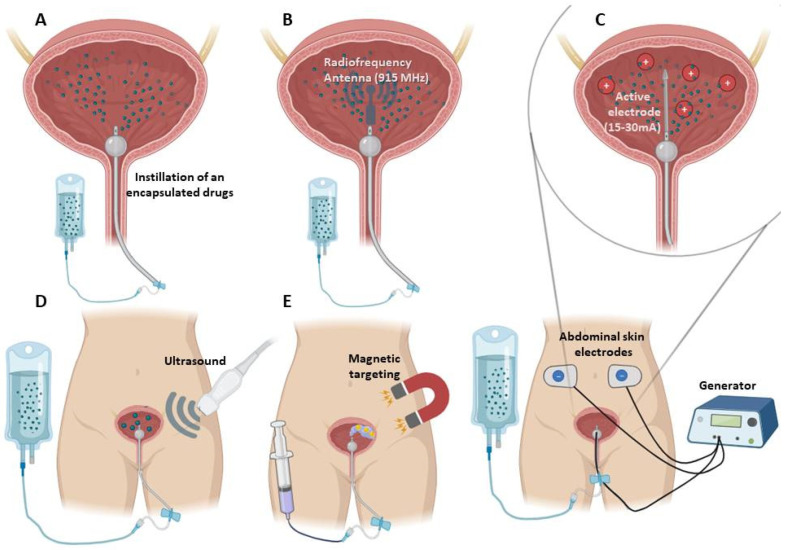
Schematic illustration of intravesical delivery of colloidal nano- and micro drug delivery systems (**A**) in combination with radiofrequency-induced hyperthermia (**B**), electromotive technology (**C**), ultrasound (**D**) and magnetic (**E**) effects. Created with BioRender.com.

## Data Availability

Not applicable.

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
