# Peer review of "Local Drug Delivery in Bladder Cancer: Advances of Nano/Micro/Macro-Scale Drug Delivery Systems"

_pharmaceutics, 2023, doi:10.3390/pharmaceutics15122724_

Round 1
Reviewer 1 Report
Comments and Suggestions for Authors
This manuscript systematically summarizes the mechanisms and application challenges of two drug delivery systems for in vivo treatment of bladder cancer (including colloidal nano- and micro-sized drug carriers, and in- 21 dwelling macro-scale devices). This review generally fulfills the requirements, both in terms of novelty, content and writing, but minor revision is required before this manuscript could be further considered. The comments listed below need to be addressed.
Comment 1: Line 72, in the introduction section, the authors stated "…Most of the work devoted to the development of new drugs for bladder cancer which are carried out in oral form…" It would be helpful to provide some examples of drugs in oral form for treatment of bladder cancer.
Comment 2: Line 113, in the introduction section, the authors stated "…The extensive development of nanotechnology has led to the emergence of a wide variety of nanostructured systems, most of which have already been tested in vitro many times…" It would be helpful to provide some references supporting this statement.
Comment 3: Please check whether Figure 3A is correct, it seems some information was missing, such as why the left four figures were different, and which differences are there in the right two figures?
Comment 4: Line 371, "…These chemicals could irreversibly disrupt the barrier function of the urothelium and, thus, might lead to multiple side effects." Some references should be added here.
Comment 5: Line 388 talks about “a Phase III randomised controlled trial did not show a difference in bladder cancer outcomes between microwave-heated chemo-therapy and standard treatment”, so is it research-worthy?
Comments on the Quality of English LanguageMinor editing of English language required
Author Response
Comment 1: Line 72, in the introduction section, the authors stated "…Most of the work devoted to the development of new drugs for bladder cancer which are carried out in oral form…" It would be helpful to provide some examples of drugs in oral form for treatment of bladder cancer.
Answer
We thank the reviewer for the valuable comment. In response, we have added information on some drugs administered orally (Lines 87-88, Refs. 6, 7, 8).
Comment 2: Line 113, in the introduction section, the authors stated "…The extensive development of nanotechnology has led to the emergence of a wide variety of nanostructured systems, most of which have already been tested in vitro many times…" It would be helpful to provide some references supporting this statement.
Answer
According to the Reviewers comment, we have added references to the articles which describe in vitro studies of different nanostructured systems on bladder cancer cells (Line 129, Refs 21-23).
Comment 3: Please check whether Figure 3A is correct, it seems some information was missing, such as why the left four figures were different, and which differences are there in the right two figures?
Answer
Regarding the floating gel example depicted in Figure 3A, the first three photos display the process of injecting a liquid mixture composed mostly of Poloxamer 407 (35%) with the addition of NaHCO3 (8%) and hydroxypropyl methyl cellulose (5%) into a citric buffer (at pH 5.0). This process simulates the introduction of a mixture into the bladder using a catheter in vivo. The red colour of the mixture results from adriamycin encapsulated in albumin nanoparticles, which is one of the components of the mixture. As can be seen from photos 1-3, this liquid mixture is forming a hydrogel immediately. In the third photograph, the catheter used to introduce the mixture is removed and the supply of the mixture is stopped. In an acidic environment such as citric acid buffer and acidified urine, the NaHCO3 decomposes and produces many CO2 bubbles (Photo 4). Floating in the citric buffer, the coloured hydrogel is aided by multiplying microbubbles that are generated both on its surface and within. The photos numbered 5-6 demonstrate the process of the gel floating, revealing the gel's ability to float in the bladder quickly after injection without causing any obstructions.
To clarify the mechanism of action of this delivery system, we have appended further elaborations to the relevant paragraph (Lines 581-591)
Comment 4: Line 371, "…These chemicals could irreversibly disrupt the barrier function of the urothelium and, thus, might lead to multiple side effects." Some references should be added here.
Answer
We thank the reviewer for the valuable comment. According to it we added references concerning possible damage of the urothelium by ethanol and acetone (lines 386-388, Refs 80, 81, 82).
Comment 5: Line 388 talks about “a Phase III randomised controlled trial did not show a difference in bladder cancer outcomes between microwave-heated chemo-therapy and standard treatment”, so is it research-worthy?
Answer
Currently, it is believed that microwave-heated chemotherapy does not enhance therapeutic effectiveness compared to standard treatment, with respect to both disease-free survival time and side effects. But it should be noted, that papillary bladder lesions may benefit from microwave-heated chemotherapy treatment; however, more research is needed. The aforementioned conclusion is based on the outcomes of second-line therapy for the treatment of patients with recurrent bladder cancer. Using a diverse sample of patients would probably result in contrasting findings.
Reviewer 2 Report
Comments and Suggestions for Authors
In this review, authors give an overview of current and future prospects of colloidal nano- and micro-sized drug delivery systems, as well as indwelling three-dimensional depot systems for bladder cancer treatment. Novel drug delivery systems could increase the residence time of the drug in the bladder and have a potential to improve the efficacy of the therapy, and therefore survival rate. Using local therapy methods to treat bladder cancer is a promising way. This review is worth of reading and can be accepted for publication in this journal.
1. As a comprehensive review, authors are strongly recommend to give a scheme and table to showcase the main content of the review, and summarize the contents that are not needed to be discussed in details.
2. From the view of materials, the review is lack of scientific contents. For example, in section 2 (Colloidal nano- and micro-sized delivery systems); in section 3 (Reservoir-type intravesical delivery systems). However, the attached figures have nothing to do with these colloidal nano- and micro-sized delivery systems, and reservoir-type intravesical delivery systems. Therefore, I suggest the authors to revise the manuscript accordingly, to show more examples on nano- and micro-sized delivery systems.
3. Related papers on the same topic can be cited: https://doi.org/10.3390/molecules28114498; https://doi.org/10.1016/j.ejps.2021.105885.
Author Response
- As a comprehensive review, authors are strongly recommend to give a scheme and table to showcase the main content of the review, and summarize the contents that are not needed to be discussed in details.
Answer
We are very grateful for the Reviewers comment. In accordance with it, we added a Table of Contents before the main material.
- From the view of materials, the review is lack of scientific contents. For example, in section 2 (Colloidal nano- and micro-sized delivery systems); in section 3 (Reservoir-type intravesical delivery systems). However, the attached figures have nothing to do with these colloidal nano- and micro-sized delivery systems, and reservoir-type intravesical delivery systems. Therefore, I suggest the authors to revise the manuscript accordingly, to show more examples on nano- and micro-sized delivery systems.
Answer
Regarding the figures, all of them except for Figure 1 display distinct delivery systems. Figure 1 relates to the introduction and portrays the various stages of bladder cancer, along with a magnified view of the bladder wall and its constituents. Figure 2 is relevant to the section titled "Colloidal nano- and micro-sized delivery systems," and it exemplifies the intravesical administration of colloidal nano- and micro drug delivery systems with diverse physical impacts. Figures 3-5 pertain to the "Reservoir-type intravesical delivery systems". Figure 3 depicts a hydrogel system, while Figures 4 and 5 illustrate various implantable devices denoting diverse types of reservoir-type intravesical delivery systems.
- Related papers on the same topic can be cited: https://doi.org/10.3390/molecules28114498; https://doi.org/10.1016/j.ejps.2021.105885.
Answer
We thank the reviewer for the comment. The mentioned article (https://doi.org/10.1016/j.ejps.2021.105885) is already cited in our review. Please find ref [119], line 580. The second article https://doi.org/10.3390/molecules28114498 does not cover the delivery of drugs for bladder cancer.
Reviewer 3 Report
Comments and Suggestions for Authors
The work concerns an important area of ​​research, which is the search for new anticancer therapies. The introduction of the work contains basic information that can be found in histology textbooks, and in my opinion, they are not necessary to understand the subject of the work. The translation of the abbreviation BCG is unclear. The high biocompatibility of chitostan should be emphasized. In the description of possible new therapies being developed, the authors limited themselves to providing a list of technologies being developed, omitting possible and already described molecular mechanisms of signal transduction. The possibility of using magnetorheological elastomers (MRE) should be taken into account.
Author Response
We thank the Reviewer for important remarks!
We have provided elemental details about the structure and physiology of the bladder in the outset of this article since it is fundamental to the scrutiny of the proposed local drug delivery vehicles. The constraints of treating bladder cancer are mainly associated with the physiological traits of the organ. Therefore, we deem this material essential in our review.
Conventional treatment for NMIBC include 1) transurethral resection of bladder tumor (TURBT), in which malignant tissue is removed with an electrocautery device during cystoscopy; 2) intravesical immunotherapy with Bacillus Calmette Guérin (BCG) vaccine or chemotherapeutic drugs, and surveillance. BCG is the Bacillus Calmette Guérin vaccine, the decoding of this abbreviation appears for the first time on line 97.
In line with the comment on the chitosan biosafety, we have added information on the biocompatibility of chitosan with reference to a study showing no toxicity to normal urothelium (Line 181, Ref 28).
The review aims to present a contemporary comprehending of local delivery methods for the management of bladder cancer. In particular, the review analyses systems of varying dimensions (nano-, micro-, macro-) that have undergone testing in vivo or are presently undergoing clinical trials. The molecular characteristics of the disease and its management lie outside the scope of our examination.
During the literature review process, no relevant information was found regarding the use of magnetorheological elastomers in the creation of treatment systems for bladder cancer. It is possible that these materials have been successfully implemented in other fields, yet their applicability to our review proves complicated.